# AdRoit is an accurate and robust method to infer complex transcriptome composition

Tao Yang[1,3], Nicole Alessandri-Haber[1], Wen Fury [1], Michael Schaner[1], Robert Breese[1], Michael LaCroix-Fralish[2], Jinrang Kim[1], Christina Adler[1], Lynn E. Macdonald[1], Gurinder S. Atwal[1] & Yu Bai [1,3]✉

Bulk RNA sequencing provides the opportunity to understand biology at the whole transcriptome level without the prohibitive cost of single cell profiling. Advances in spatial transcriptomics enable to dissect tissue organization and function by genome-wide gene expressions. However, the readout of both technologies is the overall gene expression across potentially many cell types without directly providing the information of cell type constitution. Although several in-silico approaches have been proposed to deconvolute RNA-Seq data composed of multiple cell types, many suffer a deterioration of performance in complex tissues. Here we present AdRoit, an accurate and robust method to infer the cell composition from transcriptome data of mixed cell types. AdRoit uses gene expression profiles obtained from single cell RNA sequencing as a reference. It employs an adaptive learning approach to alleviate the sequencing technique difference between the single cell and the bulk (or spatial) transcriptome data, enhancing cross-platform readout comparability. Our systematic benchmarking and applications, which include deconvoluting complex mixtures that encompass 30 cell types, demonstrate its preferable sensitivity and specificity compared to many existing methods as well as its utilities. In addition, AdRoit is computationally efficient and runs orders of magnitude faster than most methods.

[1] Regeneron Pharmaceuticals, Inc., Tarrytown, NY 10591, USA. [2] Cellular Longevity, Inc., San Francisco, CA 94103, USA. [3] These authors contributed equally: Tao Yang, Yu Bai. ✉email: yu.bai@regeneron.com

RNA sequencing is a powerful tool to understand the biology of normal and disease tissues at the whole transcriptome level. It helps to elucidate disease mechanisms and identify potential treatment targets[1]. Due to the presence of heterogeneous cell populations, the bulk tissue transcriptome only characterizes the overall gene expression across multiple cell types. The cell type identities and their prevalence remain unknown in the bulk data. However, knowledge of the cell type composition is often critical to understanding the biology. For instance, the constitution of stromal and immune cells sculpts the tumor microenvironment, which is essential in cancer progression or response to immune checkpoint inhibition[2–6]. Excessive expression of cytokines in particular leukocyte types underlines the etiology of many chronic inflammatory diseases[7–11]. Such information cannot be directly read out from the bulk RNA-Seq.

Recent breakthroughs in spatial transcriptomics methods enable characterizing whole transcriptome-wise gene expressions at spatially resolved locations in a tissue section[12]. However, it remains challenging to reach a single-cell resolution while measuring tens of thousands of genes transcriptome-wise. Some widely used technologies can achieve a resolution of 50–100 μm, equivalent to 3–30 cells depending on the tissue type[12,13]. The transcripts therein may originate from more than one cell type, resulting in another form of RNA-Seq data of multiple cell types. Unlike the bulk RNA-Seq, the profiling data at each spot contains substantial dropouts as merely a few cells are sequenced, imposing additional challenges to demystify the cell type content. We refer to bulk RNA-Seq data and spatial transcriptome data at the multi-cell resolution as compound RNA-Seq data hereafter.

The rapid development of single-cell RNA-Seq (scRNA-Seq) technologies has allowed for cell-type-specific transcriptome profiling[14]. It provides the information missing from the RNA-Seq data of tissues comprised of multiple cell types (e.g., bulk RNA-Seq). Nevertheless, the technologies have low sensitivity and substantial noise due to the high dropout rate and the cell-to-cell variability. Consequently, scRNA-Seq technologies require a large number of cells (thousands to tens of thousands) to ensure statistical significance in the results. In addition, the cells must remain viable during the capture. These requirements render the scRNA-Seq technologies costly, prohibiting their application in clinical studies that involve many subjects or cannot allow real-time tissue dissociation and cell capture. Furthermore, scRNA-Seq technologies may not be well suited to characterizing cell-type proportions in solid tissues because the dissociation and capture steps may not have the same efficiency for different cell types[15–17].

Because sequencing at the single-cell level is not always feasible and also for the purpose of better interpreting many highly valuable existing bulk RNA-Seq or spatial transcriptomic data sets, in silico approaches have been developed to infer cell-type proportions from compound RNA-Seq data[18–24]. The most common strategy is to conduct a statistical inference through the maximum likelihood estimation (MLE)[25] or the maximum a posterior probability estimation (MAP)[26] on a constrained linear regression framework, wherein the unobserved mixing proportion of a finite number of cell types are part of the latent variables to be optimized[19,21–24]. The deconvolution methods are often applied to dissect the immune cell compositions in blood samples[27–31]. However, their performance in more complex tissues, such as the nervous, ocular, respiratory, or gastrointestinal organs, remains unclear. These tissues often contain many cell types ($10–10^2$) and the difference among related cells can be subtle, rendering the deconvolution a challenging task. For example, a recent study on the mouse nervous system using scRNA-Seq found more than 200 cell clusters and many are highly similar neuronal subtypes[32].

Earlier works often utilized the transcriptome profiling of the purified cell populations to estimate the gene expressions of cell type (e.g., Cibersort)[19]. More recently, acquiring cell type-specific expression from the scRNA-Seq data was shown to be an intriguing alternative[21–24]. Although it provides higher throughput by measuring multiple cell types in one experiment, profiling at the single-cell level has a high noise floor. The accuracy of deconvolution using scRNA-Seq data as the reference may be affected by the data noise if not treated properly. Moreover, the platform difference between the compound data and the single-cell data cannot be ignored.

To overcome these challenges, additional information from the data may be considered. A recent method that weighs genes according to their expression variances across samples greatly improved the accuracy[22], highlighting the importance of gene variability in inferring cell-type composition. Some other methods and applications have pointed out the importance of cell type-specific genes[24,28,31,33]. In these works, the cell type-specific expression was only used to select the input genes (e.g., markers). Nonetheless, it measures how informative a gene is in distinguishing cell types and thus can be incorporated as a part of the model. To address the platform difference between the compound data and the single-cell data, it is sometimes assumed there exists a single scaling factor or a linearly scaled bias for all genes that can be learned and corrected accordingly. This assumption does not hold, as the impact of changing platforms is different for each gene. Though learning a uniform scaling factor might correct the difference in most genes, a few genes that remain markedly biased can easily confound the estimation, especially under a linear model framework. Thus, a gene-wise correction should be considered.

In this work, we present a new deconvolution method, AdRoit, a unified framework that jointly models the gene-wise technology bias, as well as the cell type specificity and cross-sample variability of genes. The method estimates the cell type constitution in the compound RNA-Seq samples using relevant single-cell data as a training source. Genes used for deconvolution were automatically selected from the single-cell data based on their information richness. It uses an adaptively learning approach to estimate gene-wise corrections, addressing the issue that each platform may impact genes differently. AdRoit further makes use of regularization to reduce collinearity among closely related cell subtypes that are common in complex tissues. Over a set of comprehensive benchmarking data with a varying cell composition complexity, AdRoit showed superior sensitivity and specificity to other existing methods. Applications to real RNA-Seq bulk data and spatial transcriptomics data revealed strong and expected biologically relevant information. We believe AdRoit offers an accurate and robust tool for cell-type deconvolution and will enhance the value of bulk RNA-Seq and spatial transcriptome profiling.

## Results

**Overview of the AdRoit framework**. AdRoit estimates the proportions of cell types from transcriptome data of a mixed cell population including but not limited to bulk RNA-Seq and spatial transcriptome. It directly models the raw reads without normalization, preserving the difference in total amounts of RNA transcript in different cell types. The method utilizes, as a reference, the relevant pre-existing scRNA-Seq data with cell identity annotation. It selects informative genes, estimates the mean and dispersion of the expression of selected genes per cell type, and constructs a weighted regularized linear model to infer percent combinations (Fig. 1a). Because sequencing platform bias may impact genes differently[15,34,35], a uniform adjustment for all

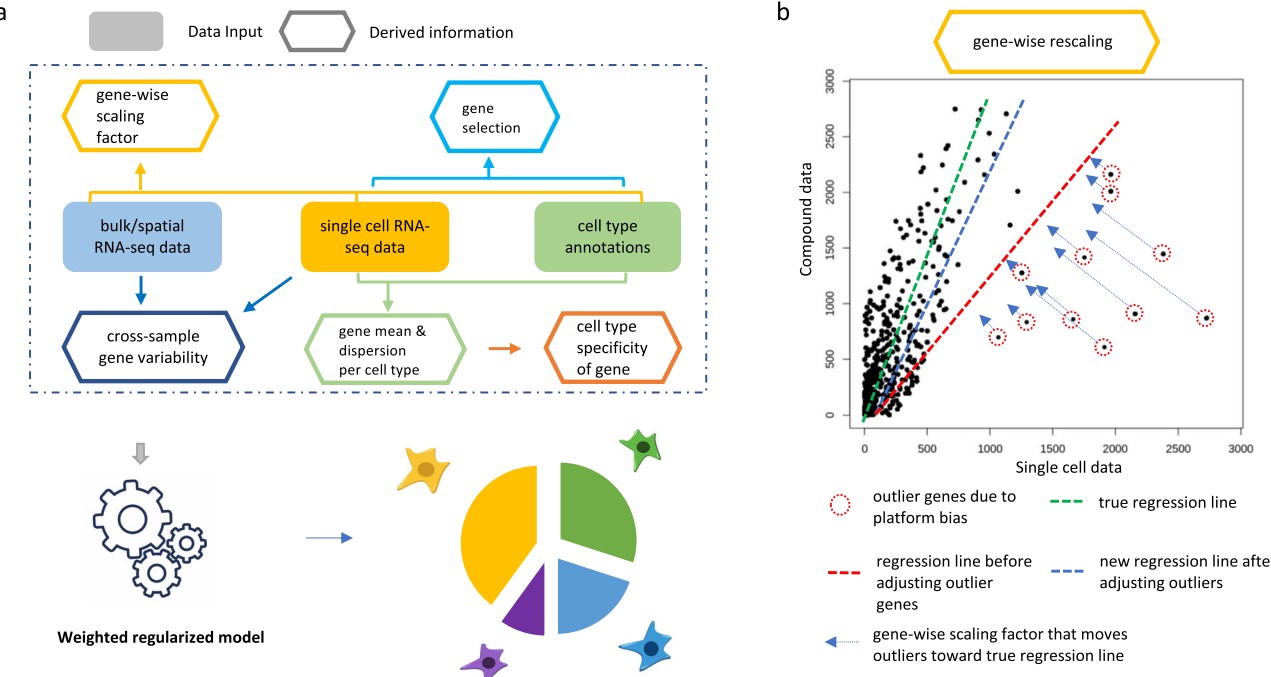

**Fig. 1 Schematic representation of AdRoit computational framework. a** AdRoit inputs compound (bulk or spatial) RNA-Seq data, single-cell RNA-Seq data, and cell type annotations. It first selects informative genes and estimates their means and dispersions, then computes the cell type specificity of genes. Depending on the availability of multiple samples, cross-sample gene variability is derived from either the compound RNA-Seq, or the single-cell data (see also "Methods"). Lastly the gene-wise correction factors are computed to reduce the platform bias between the compound and the single-cell RNA-Seq data. These quantities are used in a weighted regularized model to infer the cell type composition. **b** A mock example to illustrate the role of the gene-wise correction factor. Conceptually, an accurate estimation of the cell proportions should be represented by the slope of the green line; however, fitting in the presence of outlier genes would result in the red line. Outlier genes exist because the platform bias affects genes differently. AdRoit adopts an adaptive learning approach that first learns a coarse estimation of the slope (red line), from which the gene-wise corrections are derived and applied to the outlier genes, moving them toward the green line. The more deviated the gene, the larger the correction (i.e., longer arrows). After the adjustment, the new estimated slope (blue line) is closer to the truth (green line) and thus is a more accurate estimation.

genes may not sufficiently eliminate such bias. AdRoit adopts an adaptive learning approach, where the bias is first estimated for each gene, then applied such that more biased genes receive a larger correction (Fig. 1b).

AdRoit considers a comprehensive set of other relevant factors including the cross-sample variability and the cell type specificity of genes, as well as the collinearity of expression profiles among closely related cell types. The cross-sample variability of a gene is the variation of its expression in the same cell type across multiple bulk samples. It shall be distinguished from the expression change in the gene over different cell types. AdRoit decreases the weight of genes with high cross-sample variability whilst increasing the weight of those with an expression highly specific to certain cell types. The definition of cross-sample variability and cell type specificity also accounts for the overdispersed nature of counts data. Lastly, AdRoit adopts a linear model to ensure the interpretability of the coefficients. At the same time, AdRoit includes a regularization term to minimize the impact of statistical collinearity. Each of the factors contributes an indispensable part to AdRoit, leading to an accurate and robust deconvolution method for inferring complex cell compositions.

To evaluate the performance, we compared AdRoit with NNLS[18,36], MuSiC[22], Bisque[37], and SPOTlight[38] for bulk data deconvolution, and Stereoscope[23], Cell2location[39], and SPOTlight[38] for spatial transcriptomics data deconvolution. When evaluating the algorithms, a common practice is to pool the single-cell data to synthesize a "bulk" sample with the known ground truth of the cell composition. We measured the performance by comparing the estimated cell proportions with true proportions using four metrics:

mean absolute difference (mAD), root mean square deviation (RMSD), and two correlation statistics (i.e., Pearson and Spearman). While Pearson reflects linearity, Spearman measures whether the estimated results and the ground truth are monotonically related, even if their relationship is not linear, which avoids the artificial high linearity scores driven by outliers when the majority of estimates are small. Thus, both correlations statistics were included. Good estimations feature low mAD and RMSD along with high correlation statistics. We further applied AdRoit to real bulk RNA-Seq data and validated the results by available RNA fluorescence in-situ hybridization (RNA-FISH) data. The estimates were further confirmed by the biological knowledge of human pancreatic islets. We also used AdRoit to map cell types within spatial spots, and the accuracy was verified by in-situ hybridization (ISH) images from Allen mouse brain atlas[40].

**AdRoit excels in both simple and complex cell constitutions.**
We started with a simple human pancreatic islets data set that contains 1492 cells and four distinct endocrine cell types (Alpha, Beta, Delta, and PP cells)[41] (Supplementary Fig. 1a; Supplementary Data 1). The synthesized bulk data were constructed by mixing the single-cell data at known proportions. When using AdRoit to estimate cell proportions for a synthetic sample, data from this sample were excluded from the model construction (i.e., leave-one-out). All methods except SPOTlight achieved satisfactory performance according to the evaluation metrics, AdRoit performed the best as reflected by scatterplots of estimated proportion vs. true proportion (Supplementary Fig. 1b and

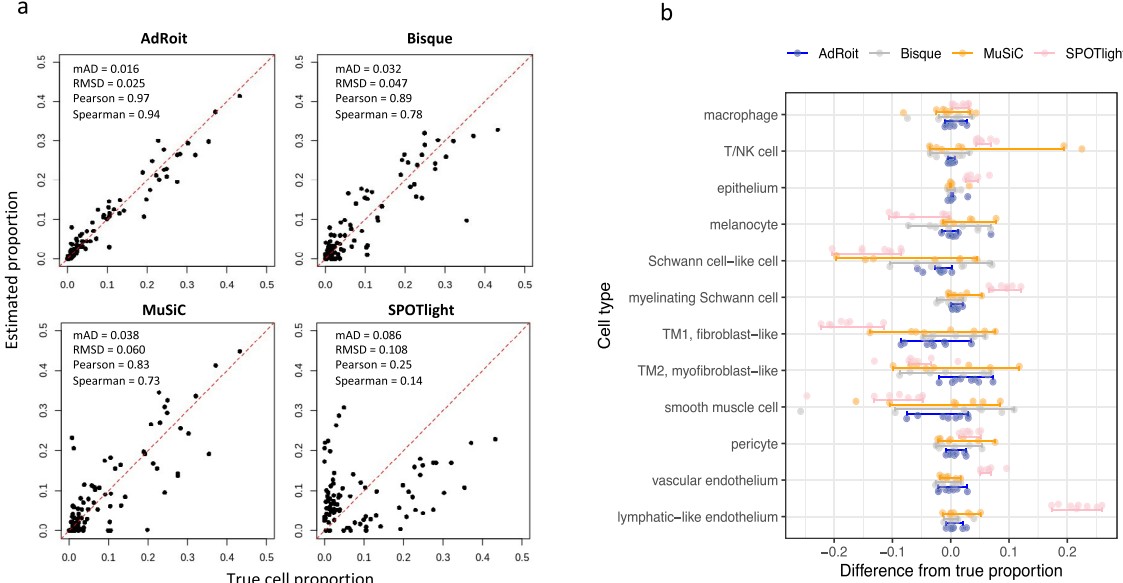

**Fig. 2 Benchmark on simulated bulk data generated from the trabecular meshwork (TM) single cells. a** AdRoit has the closest estimation to the true cell proportion comparing to Bisque, MuSiC, and SPOTlight. Each dot is a cell type from a donor. The performance metrics were derived from eight distinct donors. **b** For each cell type in TM, AdRoit has the smallest differences from the true cell type proportion and the smallest variance of estimates across eight distinct donors. For each cell type, a dot on the graph denotes a donor, and the bars represent the 1.5× interquartile ranges. The reference and gene weight estimations used for deconvoluting each synthetic bulk sample exclude the data from that sample (leave-one-out).

Supplementary Data 1). It had the lowest mAD and RMSD, and the highest correlation values (MuSiC's correlations are comparable) among the five methods tested (Supplementary Fig. 1c). Given that MuSiC is a weighted NNLS and consistently showed an improved accuracy here and in the original publication[22], we excluded NNLS from further evaluation hereafter.

We then tested the methods on a complex tissue—human trabecular meshwork (TM). We acquired published single-cell data that contains 8758 cells and 12 cell types from eight donors[42]. The data include three similar types of endothelial cells, 2 types of Schwann cells, and 2 types of TM cells (Supplementary Fig. 2; Supplementary Data 2). Cells from each donor were pooled as individual synthetic bulk samples. The cell-type proportions vary from <1 to 43%. These proportions were the ground truth cell composition and were compared head-to-head with the estimated proportions inferred by four methods. For each synthetic bulk sample, estimations by AdRoit were performed using the reference and gene weights built upon the remaining samples (i.e., leave-one-out). In each of the eight samples, the estimates made by AdRoit best approximated the true proportions. In particular, AdRoit had the lowest mAD and RMSD, and the highest correlation values among the methods (Fig. 2a). We further assessed the deviation of the estimates from the true proportions for each cell type. AdRoit consistently had the lowest deviations from the true proportions for all cell types, as well as the lowest variation among eight samples (Fig. 2b, blue dots), indicating robustness over various cell types and samples (Supplementary Fig. 3 and Supplementary Data 2).

**AdRoit has better sensitivity and specificity.** To assess the level of granularity that AdRoit can achieve when deconvoluting similar cell types, we used well-studied, closely related immune cells from myeloid lineage and lymphoid lineage. We obtained B cells, naïve $CD4^+$ T cells, memory $CD4^+$ T cells, $CD8^+$ T cells, natural killer (NK) cells, dendritic cells (DC), $CD14^+$ monocytes, $FCGR3A^+$ monocytes, and platelets from a public release of the human peripheral blood mononuclear cell (PBMC) single cells by 10x Genomics (see also "Data Availability"). We simulated bulk

samples that each contained a mixture of myeloid lineage or lymphoid lineage cell types. Such a sample was created by randomly selecting cells from a given set of cell types and mixing them according to their predefined percentages. We repeated this procedure 100 times for a series of mixing proportions among myeloid or T lymphoid cells. The bulk mixtures of $CD14^+$ monocytes, $FCGR3A^+$ monocytes and dendritic cells (DC) followed three schemes of proportions: 0.33:0.33:0.33 (mix0), 0.1:0.45:0.45 (mix1) and 0.1:0.3:0.6 (mix2). The same ratios were applied to the mixtures of naïve $CD4^+$ T, memory $CD4^+$ T, and $CD8^+$ cells. All the nine cell types in the PBMC data were used to build the reference for the deconvolution. AdRoit accurately estimated the known percentages in all cases (Fig. 3a and Supplementary Data 3). In addition, there were no noticeable false positive predictions of any excluded cell types. These results highlight that AdRoit can capture the fine difference among similar cell types and quantitatively distinguish them.

We next systematically benchmarked the sensitivity and specificity of each of the algorithms. In the context of the cell type deconvolution, a false negative occurs when the proportion of an existing cell type is predicted to be zero (or below a given threshold). Conversely, a non-zero prediction (or above a given threshold) of an absent cell type results in a false positive. False negatives and false positives measure the sensitivity and specificity of a deconvolution algorithm, respectively. Both quantities are crucial to establish the utility of the algorithm. Particularly, in real-world applications, it is often difficult to know *a prior* what cell types exist in a bulk sample, users may inform the algorithm to consider more possible cell types than what actually exists in the sample. False-positive predictions in this situation would make the algorithm unusable.

We designed a simulation to test sensitivity and specificity. we selected 6 out of the 12 human trabecular meshwork cell types, i.e., Schwann-cell-like cell, TM1, smooth muscle cell, melanocyte, macrophage, and pericyte, from each donor sample and pooled them within that sample to synthesize eight new bulk samples (Supplementary Data 4). The unselected six cell types were considered absent in the bulk samples. Some cell types selected

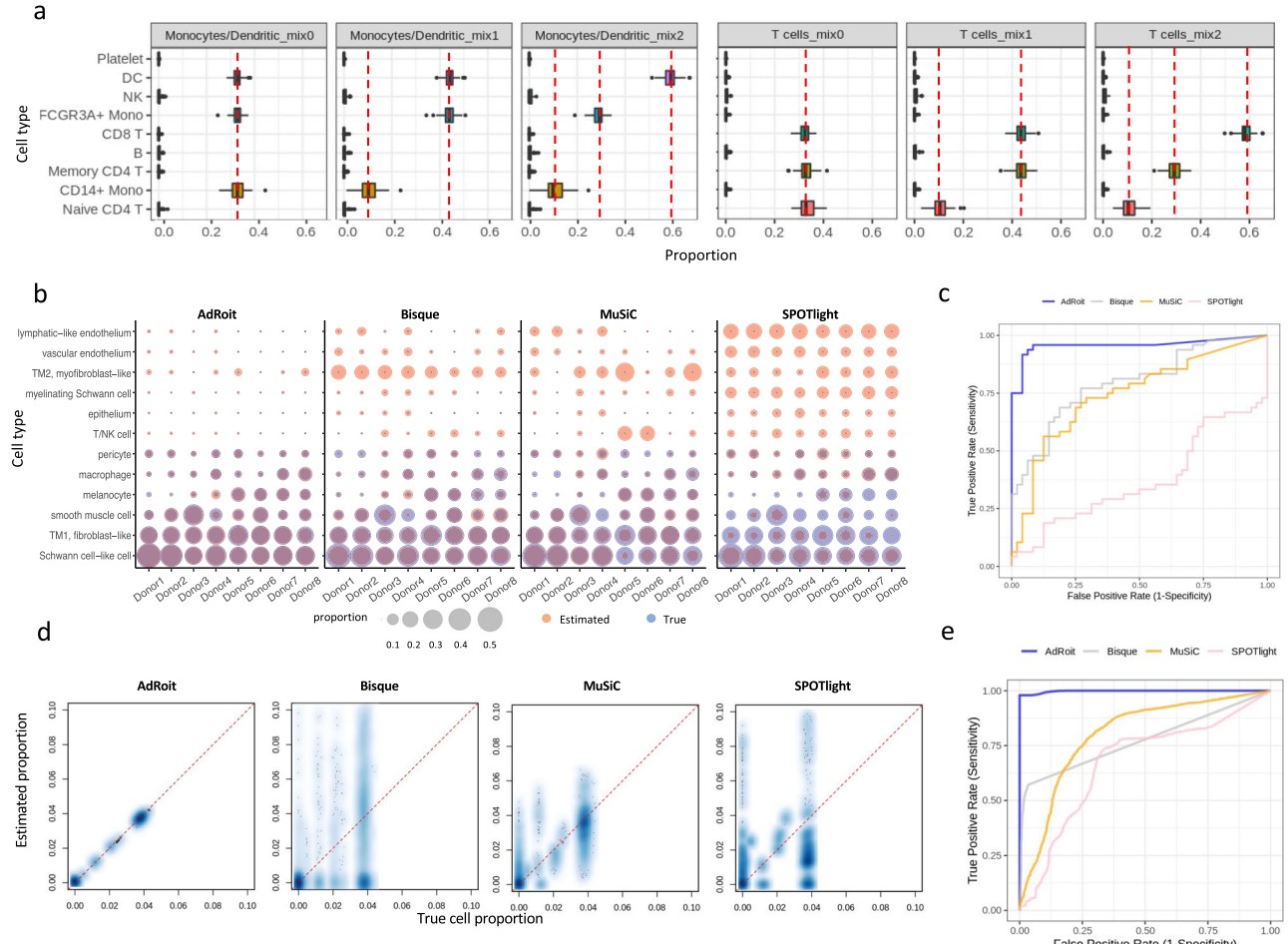

**Fig. 3 AdRoit can achieve a high granularity and exhibits good sensitivity and specificity in complex tissues. a** AdRoit is accurate in deconvoluting the simulated bulk samples that contain a mixture of similar cell types from myeloid or lymphoid lineage. The vertical dashed lines indicate the true mixing proportions. *CD14*[+] monocytes, *FCGR3A*[+] monocytes and dendritic cells (DC) were mixed under three schemes of proportions: 0.33:0.33:0.33 (mix0), 0.1:0.45:0.45 (mix1) and 0.1:0.3:0.6 (mix2). The same ratios were applied to the mixtures of naïve *CD4*[+] T, memory *CD4*[+] T, and *CD8*[+] cells. Each boxplot was derived based on $n = 100$ independent simulations, with bars denoting the 1.5× interquartile ranges. **b** AdRoit's estimates are more accurate and specific than those from Bisque, MuSiC, and SPOTlight on synthetic samples that contain only 6 out of the 12 cell types. The deconvolution was done using all 12 cell types as the reference. A pair of size-matched blue (true value) and red (estimated value) bubbles indicate an accurate prediction. Red-only and blue-only bubbles mark false positives and false negatives, respectively. **c** The comparison of Receiver operating characteristic (ROC) curves ($n = 8$ independent donors) shows that AdRoit has a notable higher area under the curve (AUC) than other methods, meaning better sensitivity and specificity. **d** Scatterplots between the ground truth and the deconvoluted cell proportions in the simulated bulk samples of high complexity (mixtures of 30 cell types). **e** ROC curves ($n = 100$ independent simulations) show AdRoit has the best AUC among all methods on highly complex cell constitutions.

highly resembled those in absence, challenging the programs to pinpoint the right cell type present in the bulk among similar candidates. We provided the full list of 12 single cell types as the reference to the programs to estimate the cell type proportions. Consistently across eight samples, AdRoit had the most accurate estimates for the six present cell types, and zero or close to zero estimated values for the non-existing cell types in the simulated bulk data. MuSiC, Bisque, and SPOTlight were notably less accurate on the six selected cell types, as well as they had many non-negligible values for the six cell types excluded in the eight synthetic samples (Fig. 3b and Supplementary Data 4). For example, smooth muscle cells accounted for ~14% in donor 4 but were largely missed (~0.03%) by MuSiC. We noted that TM2 had false non-zero estimates from all methods though not included. This is because TM2 is easily mistaken as TM1 due to their high similarity[42]. Nonetheless, AdRoit's estimates of TM2 were consistently low across samples (<1% for 44 out of 48 estimates), while the other methods had markedly larger estimates of TM2 that occasionally even exceeded the TM1 estimates. For a

systematic comparison, we constructed the receiver operating characteristic (ROC) curve by varying the threshold of detection (i.e., a cutoff below which the cell type was deemed undetected) (Fig. 3c). AdRoit had a higher area under the curve (AUC) than the other methods (AdRoit: 0.95, Bisque: 0.79, MuSiC: 0.74, SPOTlight: 0.37), implying both better sensitivity and specificity.

As complex tissues often contain several tens of different cell types, we continued the evaluation of the sensitivity and specificity in more complicated cell mixtures. We utilized the published mouse brain single-cell atlas by Zeisel et al. that contains a comprehensive set of neuronal and supporting cells[32]. Without loss of generality, we consolidated the original annotation into 46 major cell types ("Methods" and Supplementary Data 5). To synthesize a bulk mixture, we randomly selected 30 cell types and pooled all their associated cells. This procedure was repeated independently 100 times for thorough coverage of different cell type combinations. The deconvolution was performed using all 46 cell types as the reference. As illustrated in Fig. 3d, AdRoit was able to estimate the proportions closely

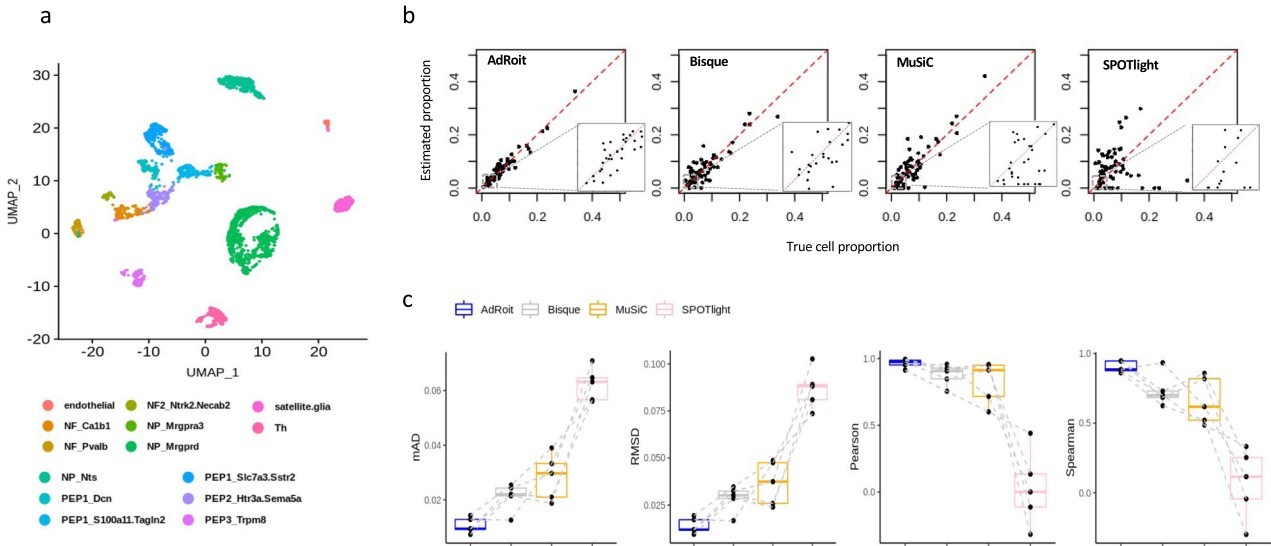

**Fig. 4 Benchmark on simulated bulk data generated using mouse dorsal root ganglion (DRG) cells containing closely related subtypes of neurons. a** 14 cell types are identified from scRNA-Seq samples of 5 mice, including multiple subtypes of neurofilaments (NF), peptidergic (PEP), and non-peptidergic (NP) neurons. **b** Benchmarking with the synthetic data shows the cell type proportions inferred by AdRoit are more accurate. In particular, AdRoit remains a better accuracy when the cells are rare (e.g., <5%; see also the zoom-in inserts). Each dot represents a cell type from one sample. **c** For each sample, mAD, RMSD, Pearson, and Spearman correlations are compared across four methods. AdRoit has the lowest mAD and RMSD, and the highest Pearson and Spearman correlations. In addition, AdRoit's estimation is the most stable across samples. Each boxplot was generated based on $n = 5$ distinct mice (one dot represents one animal). The bar of each boxplot indicates the 1.5× interquartile range. Same animals are chained by the dotted lines across the methods. The deconvolution was done by using the leave-one-out strategy.

consistent with the ground truth while the other three tools made less satisfactory predictions (see also Supplementary Data 6). The ROC curve further indicates that AdRoit has the best sensitivity and specificity given its highest AUC (Fig. 3e). These observations convince us that AdRoit offers a leading performance on highly complex cell constitutions.

**AdRoit outperforms in deconvoluting closely related cell subtypes.** To further evaluate AdRoit when multiple homologous subtypes of cells are present in a complex tissue, we performed scRNA-Seq experiment on mouse lumbar dorsal root ganglion (DRG) from five mice. Following the standard analysis pipeline (Methods), we obtained 3352 single cells after quality control procedures. After clustering and annotation, we discovered 14 cell types including multiple subtypes of neuronal cells (Fig. 4a and Supplementary Data 7). The heatmap of the top marker genes showed distinct patterns of the major cell types as well as similar patterns of the subtypes (Supplementary Fig. 4a), with the cell type proportions varying from 0.5 to 33.71% (Supplementary Fig. 4b). These 14 cell types include three subtypes of neurofilament containing neurons (i.e., NF_Calb1, NF_Pvalb, NF_Ntrk2.Necab2), three subtypes of non-peptidergic neurons (i.e., NP_Nts, NP_Mrgpra3, NP_Mrgprd), and five subtypes of peptidergic neurons (i.e., PEP1_Dcn, PEP1_S100a11.Tagln2, PEP1_Slc7a3.Sstr2, PEP2_Htr3a.Sema5a, PEP3_Trpm8). Also discovered were tyrosine hydroxylase-containing neurons (Th), satellite glia, and endothelial cells. Such complex compositions formed a challenging testing ground for evaluating the ability to distinguish closely related cell types. We again performed the leave-one-out deconvolution on five synthesized bulk samples.

AdRoit had highly accurate estimations on all cell subtypes across samples (Fig. 4b). Worth noting, for the rare cell types that account for less than 5%, AdRoit still had a good estimation that is close to the true proportions and did not miss a cell type, showing that AdRoit is robust on rare cell types (Supplementary Fig. 5 and Supplementary Data 7). Conversely, Bisque, MuSiC, and

SPOTlight were notably less accurate, especially for the cell types less than 5%, and all three missed multiple cell types including some large clusters accounting for ~10% of the cells (PEP1_Slc7a3.Sstr2 cells of Sample5). We continued to examine how much the variability of the estimates was in each sample. We computed the same four metrics used previously to evaluate the performance on each of the five synthetic samples and compared them head-to-head among the algorithms. This fine comparison showed AdRoit notably outperformed on nearly every sample (Fig. 4c). Moreover, the performance metrics of AdRoit were highly consistent across samples with the lowest variability among all four methods.

**AdRoit consistently performs well on spatial transcriptomics data.** Given the promising performance on complex tissues, we moved forward to test AdRoit's applicability to spatial transcriptome data. Spatial transcriptomics data differs from bulk RNA-Seq data in that each spot only contains transcripts from a handful of cells (3–30)[12]. Some of the spots contain multiple cells of the same type, while others may have heterogeneous cell types at varying mixing percentages (e.g., spatial spots at the boundary of different cell types). As the spot is a pool of only a few cells, the variations across spatial spots are expected to be greater than the changes in bulk samples, imposing an additional challenge for deconvolution. We simulated a large number of spatial spots (2900 in total) by using sampled cells from the DRG single-cell data above (Methods), then compared AdRoit with Stereoscope, Cell2location, and SPOTlight over a range of simulation scenarios.

We first tested whether the methods could correctly infer a single cell type when the spots contain cells from that same type. For each of the 14 cell types from DRG, we sampled 10 cells and pooled them to form a spatial spot. We repeated the simulation 100 times for robust testing, then used the full set of 14 cell types as a reference to deconvolute the 1400 simulated spots. All methods except SPOTlight were able to identify the correct cell types with high accuracy on the target cell types (i.e., percentages

close to 1) and comparably low estimated values (i.e., percentages close to 0) for other cell types excluded from the simulation (Supplementary Fig. 6 and Supplementary Data 8).

We then proceeded to a more difficult scenario where we sampled cells from the 5 PEP neuron subtypes and mixed them. We created three simulation schemes for a comprehensive evaluation: (1) 5 PEP subtypes had the same percent of 0.2; (2) PEP1_*Dcn* was 0.1 and the other 4 were 0.225; (3) PEP1_*S100a11.Tagln2* and PEPE1_*Dcn* were 0.1, PEP2_*Htr3a.Sema5a* and PEP1_*Slc7a3.Sstr2* were 0.2, and PEP3_*Trpm8* was 0.4. Again, each simulation scheme was repeated 100 times. Under each scheme, the estimates by AdRoit consistently centered around the true proportions while the predictions of irrelevant cell types remained close to 0 (Fig. 5a and Supplementary Data 8). In comparison, Cell2location systematically overestimated the PEP1_*Slc7a3.Sstr2* by about 2.5%. Stereoscope more notably overestimated PEP2_*Htr3a.Sema5a* and underestimated the PEP3_*Trpm8*. SPLOTlight had high noise for some cell types (e.g., endothelial and satellite glia) and generally deviated from the truth for multiple cell types under all simulation schemes.

We expanded the simulated spatial spots to the mixture of 3 NP cell subtypes and the mixture of 3 NF subtypes. In addition, we blended either NP_*Mrgpra3* cells, NF_*Calb1* cells, or PEP3_*Trpm8* cells with three other cell types (i.e., Th, satellite glia, and endothelial). For all these simulated spatial spots, AdRoit's estimates consistently centered around the true proportions, whereas the other methods deviated by different degrees in most of the simulated schemes (Supplementary Fig. 7 and Supplementary Data 8). We speculate one important reason for the underperformance of some methods such as Stereoscope is that they normalize the total UMI counts to the same value for all cells. However, in real world, different cell types are unlikely to have the same amount of RNA transcripts (e.g., immune cells have about 10 fold fewer total UMIs than the neurons). Our simulation pooled the cells by adding up the raw UMI counts per gene, which we believe best mimics the real data. Normalizing the data by total UMI not only eliminates this factual difference between cell types but may distort the data from the assumed negative binomial distribution. AdRoit avoids this problem by model the raw UMI counts.

Next, we asked how sensitive the methods are in detecting rare cell populations. We simulated mixtures of 3 PEP subtypes (i.e., PEP1_*Slc7a3.Sstr2*, PEP2_*Htr3a.Sema5a*, and PEP3_*Trpm8*) wherein the percentage of PEP3_*Trpm8* cells varied from 0.01 to 0.1 by 0.01, and the other two cell types sharing the remaining percentage equally (Methods). At each given mixing ratio, the simulation was repeated 100 times. We then checked how accurately the concentration of PEP3_*Trpm8* cells was estimated. The medians of AdRoit's estimates were closest to the true proportions (Fig. 5b, red dashed lines) among all four methods, followed by Cell2location. Stereoscope's estimates were systematically lower than the true values and failed to detect the PEP3_*Trpm8* cells when the simulated proportion was below 0.06. On the other hand, SPOTlight constantly overestimated the presence of PEP3_*Trpm8* cells. The median of its predictions remained around 9% even when the true population was at 1%. Such observations are consistent with the frequent false positives made by SPOTlight when deconvoluting the simulated bulk samples (Fig. 3b). This comparison implied AdRoit is more advantageous in detecting low percent cells. For a complete evaluation, we replicated the comparison using five additional sets of cell mixtures: NF_*Calb1* with NF_*Pvalb* and NF2_*Ntrk2.Necab2* (NF subtypes); NP_*Mrgpra3* with NP_*Mrgprd* and NP_*Nts* (NP subtypes); NF_*Calb1* with Th, satellite glia and endothelial (NF_*Calb1* + others); NP_*Mrgpra3* with Th, satellite glia and endothelial (NP_*Mrgpra3* + others); and PEP_*Trpm8* with Th,

satellite glia and endothelial (PEP_*Trpm8* + others). In each set, the first cell type mentioned was considered rare with its percent varied from 0.01 to 0.1. The rest cell types shared the remaining proportion evenly. At each given percent of the rare cell type, we computed how many times out of 100 the rare cell component was detected (estimation > 0.005). AdRoit had systematically high detection rates especially when the percent is below 3% (Fig. 5c and Supplementary Data 9). Note that the apparent high detection rates in SPOTlight were merely the result of its high false-positive estimates (Figs. 3b and 5b). The detection sensitivity of Cell2location was comparable to that of AdRoit. Notably, given a rare cell population of 5%, both AdRoit and Cell2location achieved a detection rate >90% under all simulation schemes, making them powerful tools to uncover rare cells.

**Application to real bulk RNA-Seq data of human pancreatic islets.** Though using synthetic bulk data based on the mixing of single cells is a useful benchmarking strategy, the bulk and single-cell RNA-Seq often use distinct RNA library preparations and sequencing protocols. The capability of a method to deconvolute real bulk samples shall be addressed to ensure it is useful in real-world applications. We acquired 70 real human pancreatic islets bulk samples from published studies[41,43,44] (Supplementary Data 10) and used single-cell data of the same tissue[41] as the reference to infer the percentages of four endocrine cell types (i.e., Alpha, Beta, Delta, PP). The 70 bulk samples were collected from 39 distinct donors, including 26 healthy donors, and 13 donors with type 2 diabetes (T2D). Each donor contributed 1–5 bulk RNA sample repeats.

Replicates from the same donor are expected to have similar compositions and thus were used to assess the reproducibility of the estimates from AdRoit. For all cell types, AdRoit had consistent estimates for the same donors (Fig. 6a and Supplementary Data 11). The average standard deviations did not exceed 1% for all four cell types (i.e., Alpha: 0.010; Beta: 0.008; Delta: 0.004; PP: 0.002). To seek an independent validation, we obtained cell sorting results by RNA-FISH for 4 of the 39 donors[41] (Supplementary Data 11). The estimated cell proportions in the four samples agreed well with the percentages measured by RNA-FISH (Fig. 6b). The consistency held for both major cells (Alpha and Beta) and the minor cells (Delta and PP). The observed reproducibility and the success of the independent validation showed AdRoit is reliable in deconvoluting real bulk RNA-Seq data.

We then asked if AdRoit can detect known biological differences between healthy and T2D donors. Loss of functional insulin-producing Beta cells is a prominent characteristic of T2D[45–47], typically reflected by an elevated level of hemoglobin A1c (HbA1c)[48,49]. Among the healthy donors, most Beta-cell proportions estimated by AdRoit ranged from 50 to 75% (Fig. 6c), agreeing with the known percent range of Beta cells in human islets tissue[50,51]. A significant decrease in the estimated Beta-cell proportions was seen in T2D patients (*p* value = 4.1e − 6 by two-sided t-test). Further, a linear regression of estimated Beta-cell proportions on HbA1c levels showed a statistically significant negative association (*p* value = 1.8e − 6 by two-sided t-test on the regression coefficient) showing that AdRoit adequately reflected the cell composition difference between healthy donors and T2D patients.

**Application to mouse brain spatial transcriptomics.** We lastly demonstrated an application to the real spatial transcriptomics data. Given that the molecular architecture of brain tissues has been well studied, for this evaluation, we chose a set of mouse brain spatial transcriptomics data generated by 10x genomics,

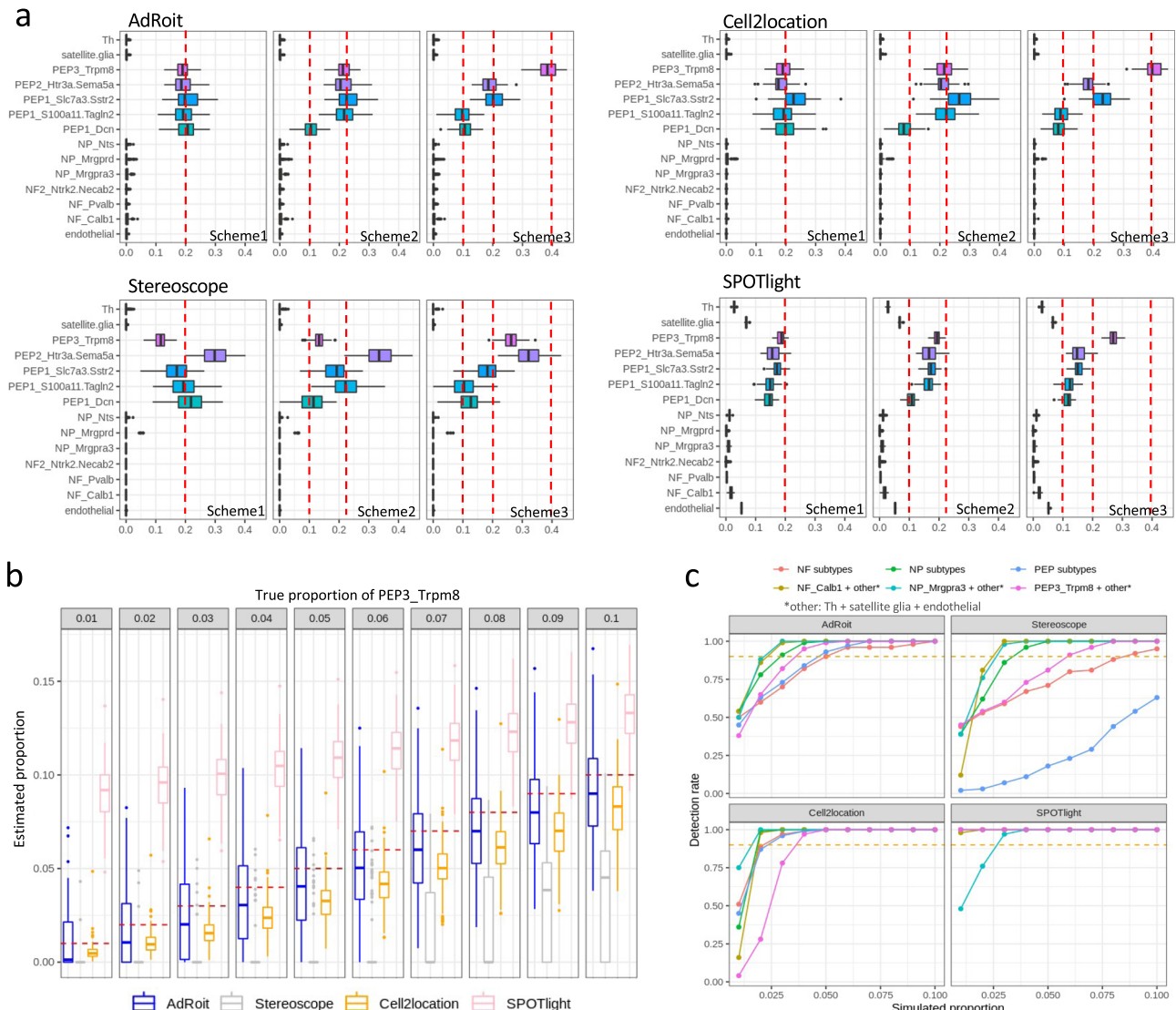

**Fig. 5 AdRoit shows a good accuracy and sensitivity in deconvoluting spatial spots simulated from dorsal root ganglion cells. a** Estimations from AdRoit, Cell2location, Stereoscope, and SPOTlight on simulated spatial spots that contain 5 PEP neuron subtypes. True mixing proportions are denoted by the red dashed lines. Three schemes are presented: (1) the proportions of 5 PEP cell types are the same and equal to 0.2; 2) PEP1_*Dcn* is 0.1 and the other 4 are 0.225; 3) PEP1_*Dcn* and PEP1_*S100a11.Tagln2* are 0.1, PEP1_*Slc7a3.Sstr2* and PEP2_*Htr3a.Sema5a* 0.2 are 0.2, and PEP3_*Trpm8* is 0.4. The boxplots were derived from $n = 100$ independent simulations. **b** The performance of AdRoit, Cell2location, Stereoscope, and SPOTlight in estimating rare cell populations in the spatial spots. The spots contain a mixture of three PEP cell subtypes (i.e., PEP1_*Slc7a3.Sstr2*, PEP2_*Htr3a.Sema5a*, and PEP3_*Trpm8*), with the percent of PEP3_*Trpm8* ranging from 1 to 10% and the other two cell types sharing the remaining proportion equally. The boxplots were drawn upon $n = 100$ independent simulations. **c** Compare the rate of detecting rare cells in simulated spots. An inferred percent greater than 0.5% is deemed as a positive detection. Six sets of cell mixtures are employed: NF_*Calb1* with NF_*Pvalb* and NF2_*Ntrk2.Necab2* (NF subtypes), NP_*Mrgpra3* with NP_*Mrgprd* and NP_*Nts* (NP subtypes), PEP3_*Trpm8* with PEP1_*Slc7a3.Sstr2* and PEP2_*Htr3a.Sema5a* (PEP subtypes), NF_*Calb1* with Th, satellite glia and endothelial (NF_*Calb1* + others), NP_*Mrgpra3* with Th, satellite glia and endothelial (NP_*Mrgpra3* + others), and PEP_*Trpm8* with Th, satellite glia and endothelial (PEP_*Trpm8* + others). In each set, the first cell type listed is the target of detection and varies its percent from 1 to 10%. The rest cell types split the remaining proportion evenly. The red dashed lines mark the detection rate of 90%. The rates were computed based on $n = 100$ independent simulations. Bars in the boxplots mark the 1.5× interquartile ranges.

containing 2703 spatial spots. The consolidated version of the aforementioned mouse brain single-cell atlas[32] was used as the reference for deconvolution ("Methods").

The cell contents inferred by AdRoit per spot appear to accurately match the expected cell types at that location (Supplementary Fig. 8 and Supplementary Data 12). For example, the three subtypes of cortex excitatory neurons each occupied a sub-area in the cerebral cortex region. As another example, the shape of the hippocampal region was delineated by the estimated percentages of dentate gyrus granule/excitatory neurons. As an

independent validation, we checked the consistency between the estimated cell types with the in-situ hybridization (ISH) images from Allen mouse brain atlas[40]. We chose four genes highly expressed in 4 brain regions respectively, i.e., *Spink8* for the hippocampal field CA1; *C1ql2* for the dentate gyrus; *Clic6* for the choroid plexus; and *Synpo2* for the thalamus[32]. The spots enriched with the four indicative cell types, hippocampal CA1 excitatory neuron type 2, dentate gyrus granule neuron type 2, choroid plexus cell, and thalamus excitatory neuron type 1, as mapped by AdRoit, precisely co-localized with the regions that

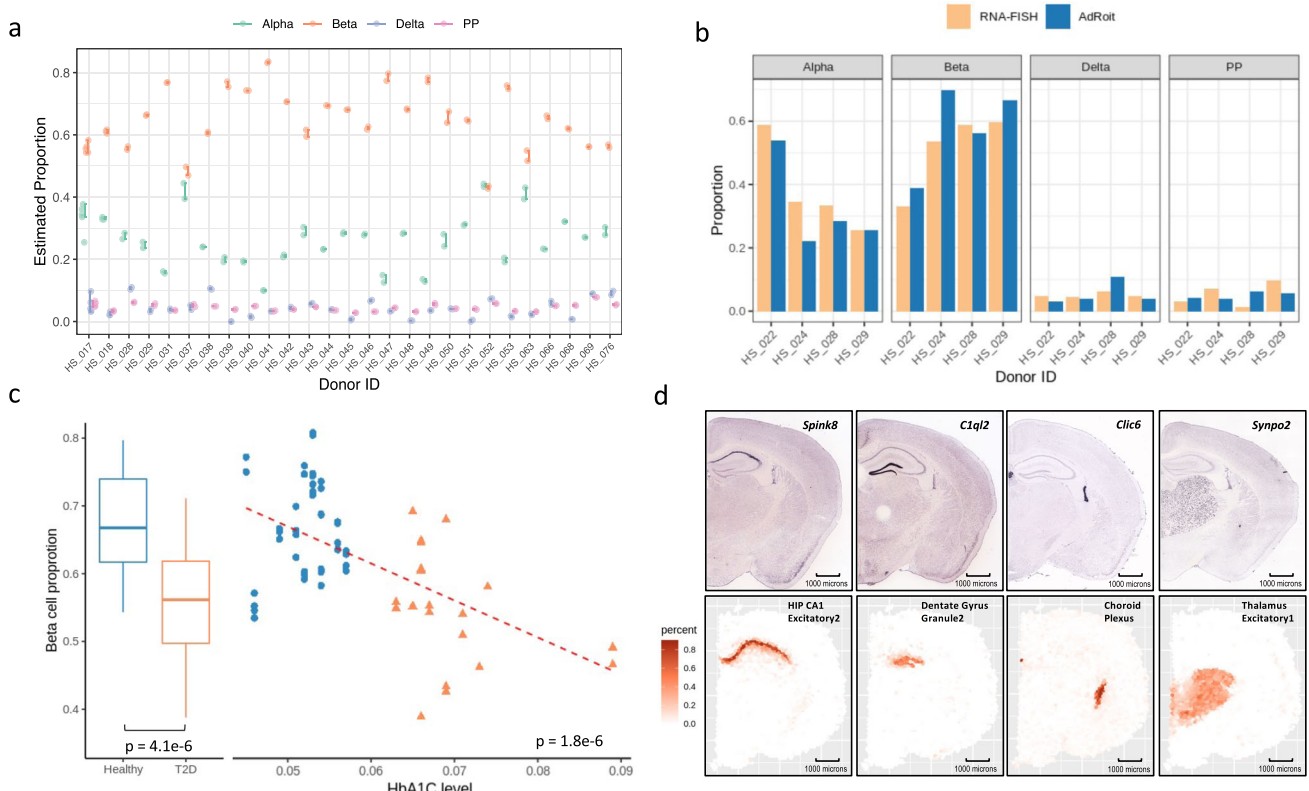

**Fig. 6 Applications to real bulk RNA-Seq data and mouse brain spatial transcriptome data. a** The deconvoluted cell compositions in the real bulk RNA-Seq data of human Islets are highly reproducible for the repeated samples from the same donor. **b** AdRoit estimation of the cell type proportions agrees with the RNA-FISH measurements. **c** AdRoit-inferred Beta-cell proportions in type 2 diabetes patients ($n = 13$ distinct subjects) are significantly lower than those in healthy subjects ($n = 26$ distinct subjects). Bars in the boxplots represent the 1.5× interquartile ranges. In addition, the estimated proportions have a significant negative linear association with the HbA1C levels ($n = 36$ distinct donors with valid HbA1C measurements). All statistical metrics were derived based on. **d** The spatial mapping of four mouse brain cell types is consistent with the locations of four region-specific markers shown on the ISH images obtained from Allen mouse brain atlas[40]. The four genes, *Spink8*, *C1ql2*, *Clic6*, and *Synpo2*, were identified by Zeisel et al.[32] as markers of the hippocampal field CA1, dentate gyrus, choroid plexus, and thalamus, respectively.

showed a strong signal of the 4 marker genes on the ISH images respectively (Fig. 6d). This agreement confirmed that the spatial mapping of cell types by AdRoit is reliable.

**Computational efficiency.** Besides the accuracy and robustness, another major advantage of AdRoit is its magnitude higher computational efficiency. AdRoit uses a two-step procedure to do the inference. The first step prepares the reference using the single-cell data where the per-gene means and dispersions are estimated, and cell-type specificity is subsequently computed. The built reference can be saved and reused. We tested the running time on the reference construction using the mouse brain single-cell dataset discussed earlier that contains ~15,000 cells. It took about 4.5 min on a CPU that has 24 cores (23 used for parallel computing). The second step inputs the built reference and the compound RNA-Seq data and does the estimation. Deconvoluting ~2700 RNA-Seq samples of mixed cell populations took around 5 min. Therefore, AdRoit in total took less than 10 min and ~3 Gb memory usage on a regular CPU. As a comparison, MuSiC and Bisque took about 1 h and 30 min on the same data using the same CPU setup. SPOTlight took about 2 h with the default parameters (cl_$n = 100$, hvg = 3000, ntop = 200, transf = "uv", method = "nsNMF", min_cont = 0). Stereoscope ran about 24 hours continuously with the published parameter setting (-scb 256 -sce 75000 -topn_genes 5000 -ste 75000 -lr 0.01 -stb 100 -scb 100) on a powerful V100 GPU with 80 cores and 16G memory. The efficiency of Cell2location was comparable to Stereoscope, taking about 18 h on the V100 GPU (posterior n_sample = 1000,

n_iteration = 10000, selection_specificity = 0.45). These methods can be prohibitive when seeking a quick turnaround.

## Discussion

In this work, we have demonstrated that AdRoit is capable of deconvoluting the cell compositions from the compound RNA-Seq data with a leading accuracy, measured by the consistency between the true and predicted cell proportions. Its advantage over the existing state-of-the-art methods was verified over a wide range of use cases. In particular, AdRoit excelled in complex tissues composed of more than ten different cell types with a wide range of cell proportions (e.g., trabecular meshwork, mouse brain, and dorsal root ganglion). In these cases, AdRoit consistently performed better than the comparators on deconvoluting bulk RNA-Seq data. AdRoit is also more accurate and sensitive than tools specifically designed for deconvoluting spatial transcriptomics spots, especially in detecting low percent rare cells. Previous benchmarking often assumed the types of cells in the synthetic bulk data are not more or less than the cell types collected in the reference, and thus the only unknown was the proportion of each cell type. This assumption may not hold. Missing existing cell types or false predictions of non-existing ones can hinder the utility of an algorithm. Thus, besides the overall accuracy, we also examined the sensitivity and specificity of the algorithms. We observed a superior sensitivity and specificity in AdRoit, an important leverage for its usage in practice.

The reference single-cell data used by AdRoit came from different platforms, such as the 10x Genomics Chromium Instrument (the mouse dorsal root ganglion), and the Fluidigm C1 system (the human pancreatic islets data). AdRoit consistently exhibited excellent performance across all benchmarking datasets independent of their single-cell sequencing technology platforms. More importantly, this statement holds not only for deconvoluting the synthesized bulk data, but also for the real bulk RNA-Seq data. The latter typically does not apply the unique molecular barcoding and requires a different cDNA amplification procedure from what is used in the single-cell RNA-Seq ("Methods"). Moreover, the sequencing depth, read mapping and gene expression quantification are dissimilar as well. The fact that AdRoit accurately dissected the cell compositions in the real bulk samples based on the single-cell reference data further supports its cross-platform applicability.

We attribute the power of AdRoit to its comprehensive modeling of relevant factors. Firstly, the impact of platform difference on genes may be different and not linearly scaled. Correcting such differences entails adjustments specifically tailored to each gene. AdRoit uses an adaptive learning approach to estimate such gene-wise correcting factors and does the correction in a unified model. In addition, the contribution of a gene in a cell type to the loss function is jointly weighted by its specificity and variability, where specificity and variability are defined in a way accounting for the overdispersion property of counts data. Our observations over the multiple benchmarking data sets also show that the coexistence of similar cell types may have induced a collinearity condition that negatively impacted the regression-based methods developed by others. Being able to alleviate this problem gives AdRoit an edge. All these factors help AdRoit to distinguish similar cell clusters while remaining sensitive enough to separate rare cell types.

Technically, the input profiles of individual cell types to AdRoit need not come from the single-cell RNA-Seq. Bulk RNA-Seq profiles of individual isolated cell types can be used as well. Nevertheless, using single-cell RNA-Seq data as the reference has a few key advantages. It is a high throughput approach wherein multiple cell types can be interrogated simultaneously. Prior knowledge of the cell types in presence, as well as their specific gene markers are not required, which allows novel cell types to be identified. Although detection of lowly expressed genes has been a challenge for the single-cell RNA-Seq, great enhancements have been demonstrated. For example, the number of detectable genes currently can reach an order of 10,000 per cell and keeps improving[52]. As AdRoit focuses on the informative genes whose expressions are generally high, the detection limit of the single-cell RNA-Seq does not impose a noticeable drawback. Indeed, given the single-cell reference profiles, AdRoit successfully deconvoluted the real bulk RNA-Seq data and spatial transcriptomics data. The results suggest that, besides enriching our understanding of the bulk transcriptome data, AdRoit can leverage the usage of the vast amount and continuously growing single-cell data as well.

AdRoit is a reference-based deconvolution algorithm. A comprehensive collection of the possible cell components is important. However, completeness may not always be guaranteed. Even with the single-cell acquisition that is independent of prior knowledge, rare and/or fragile cell types may not survive through the capture procedure and hence are excluded. It is also difficult to generate a solid reference profile for cells that are versatile from sample to sample (e.g., tumor cells). Currently AdRoit deals implicitly with the components unknown to the reference. If an unknown cell type reassembles one of the referenced ones, it may be considered as part of the known cell type and their joint population is predicted. Such an outcome is acceptable as treating two similar cell types as one is still biologically meaningful although the resolution of the system may be compromised. If the unknown component is dissimilar to all the known ones, it will be ignored by AdRoit because its representative markers are unlikely among the top-weighted genes associated with the known components. At the same time, the distinct component is expected to have a unique gene expression pattern and thus unlikely to interfere with the gene expressions from the known cell types. Therefore, AdRoit essentially deconvolutes the relative populations among the known cell components. For example, AdRoit was able to correctly uncover the populations of four endocrine cell types from the human islet bulk data despite the absence of many other cell types such as macrophages, Schwann cells, and endothelial cells in the input single-cell reference[20]. Although under such a circumstance, the absolute percentages of the cells remain obscure, we expect their relative proportions can be studied and valuable. A future improvement is to explicitly model the unknown cell types and estimate their percentages upon the signals in the compound data that cannot be explained by the contribution from the known components. AdRoit may also be coupled with other programs to output the deconvoluted cell type-specific transcriptome profiles. The inferred cell type proportions by AdRoit can serve as the input to the downstream program and benefit the outcome with the reliability and accuracy demonstrated in this work.

## Methods

**Gene selection**. AdRoit selects genes that provide information about cell type identity, excluding non-informative genes that potentially introduce noise. There are two ways for selecting such genes: (1) union of the genes whose expression is enriched in one or more cell types in the single-cell UMI count matrix. These genes are referred to as marker genes; (2) union of the genes that vary the most across all the cells in the single-cell UMI count matrix, referred to as the highly variable genes. For example, marker genes can be defined as the union of the top 200 perturbed genes that pass a p-value cutoff of 0.01, ranked by fold change, from each cell type in comparison to the rest. Considering some genes may be enriched in more than one cell type, we selected markers presenting in no more than 5 cell types to ensure specificity.

AdRoit also offers the option to use highly variable genes. To avoid the selected highly variable genes being dominated by large cell clusters, AdRoit first balances the cell types in the single-cell UMI count matrix by finding the median size (i.e., the number of cells) of all clusters, then samples cells from each cluster to make its size equal to the median. Next, AdRoit computes the variance of each gene across all the cells in the balanced single-cell UMI matrix. The variance-stabilizing transformation (VST)[53] is applied to normalize the data prior to the variance calculation. Genes with the top largest variances are then selected.

In both ways, mitochondria genes are excluded as their expression does not have information of cell identity. The results shown in the current paper were based on the marker genes derived as described above. We also demonstrated that using the balanced highly variable genes yielded comparably accurate estimations (Supplementary Fig. 9).

**Estimate gene-wise mean and variance per cell type**. Modeling single-cell RNA-Seq data is challenging due to cell heterogeneity, technical sensitivity, and noise. Some genes may not be detected by chance, while others may be found to be highly dispersed. These factors can lead to excessive variability even within the same cell type. AdRoit combats high noise and computational complexity by building models with estimated mean and dispersion per gene per cell type. This strategy reduces the data complexity while preserves the cell type-specific information.

Although typical analyses of RNA-Seq data start with normalization, AdRoit does not do normalization before the mean estimation. Performing a normalization across all cell types forces every cell type to have the same amount of RNA transcripts, measured by the total unique molecular identifier (UMI) counts per cell. However, different cell types can have dramatically different amounts of transcripts. For example, the amount of RNA transcripts in neuronal cells is about 5 times higher than that in glial cells. Thus, normalization can falsely alter the relative abundance of cell types, misleading the estimation of cell type percentages. To avoid this problem, AdRoit models the means using the raw UMI counts.

Studies have shown that UMI counts follow a negative binomial distribution[54,55]. We, therefore, fit negative binomial distributions to genes of each cell type and build the model based on the estimated means and dispersions from the selected genes. More specifically, let $X_{ik}$ be the set of single-cell UMI counts of gene $i \in 1, .., I$ for all cells in cell type $k \in 1, ..., K$. $I$ is the number of selected genes, and $K$ denotes the number of cell types in the single-cell count matrix. The distribution of $X_{ik}$ follows a negative binomial distribution,

$$X_{ik} \sim NB(\lambda_{ik}, p_{ik}), \tag{1}$$

where $\lambda_{ik}$ is the dispersion parameter of the gene $i$ in cell type $k$, and $p_{ik}$ is the

success probability, i.e., the probability of one observed UMI belonging to gene $i$ in cell type $k$. The two parameters are estimated by MLE. The likelihood function is

$$LH(\lambda_{ik}, p_{ik}|X_{ik}) = \prod_{i=1}^{n_k} f(X_{ik}|\lambda_{ik}, p_{ik}), \tag{2}$$

where $n_k$ is the number of cells in cell type $k$, and $f$ is the probability mass function of the negative binomial distribution. The MLE estimates are then given by

$$(\widehat{\lambda_{ik}}, \widehat{p_{ik}}) = arg\max_{\lambda_{ik}, p_{ik}} LH(\lambda_{ik}, p_{ik}|X_{ik}) \tag{3}$$

Once the success probability and the dispersion are estimated, the mean and the variance of the negative binomial distribution can be computed numerically,

$$\mu_{ik} = \frac{\widehat{\lambda_{ik}} \cdot \widehat{p_{ik}}}{1 - \widehat{p_{ik}}}, \tag{4}$$

$$\sigma_{ik}^2 = \frac{\widehat{\lambda_{ik}} \cdot \widehat{p_{ik}}}{(1 - \widehat{p_{ik}})^2}. \tag{5}$$

Model fitting using MLE has been readily coded in many R packages. We choose the 'fitdist' function from the 'fitdistrplus' package[56] for its fast computation speed and flexibility in selecting distributions. The mean and variance estimations are done for each selected gene in each cell type, resulting in a $I \times K$ matrix each.

**Cell type specificity of genes.** Genes with a cell-type-specific expression pattern better distinguish cell types, thus are more important for resolving cell-type composition. In line with this property, AdRoit weighs more the genes with a higher cell-type specificity. Highly specific genes usually have consistently high expression and thus relatively low variance among cells within a cell type. To compute the cell type specificity of a gene, we first identify the cell type in which the gene has the highest expression (i.e., most expressed cell type), then defines the specificity of this gene as the mean-to-variance ratio within that cell type. A high ratio renders a high weight to the gene in the model. Here the mean and the variance refer to the estimated values from the negative binomial model ($\mu_{ik}$ and $\sigma_{ik}^2$ in Eqs. (4) and (5)). Let $k'$ be the index of the cell type that has the highest mean expression of gene $i$,

$$k' = arg\max_k\{\mu_{ik}|k\epsilon 1 \dots K\}, \tag{6}$$

then the weight of the cell type specificity for gene $i$, denoted $w_i^S$, is given by,

$$w_i^S = \frac{\mu_{ik'}}{\sigma_{ik'}^2}, \tag{7}$$

and it is computed for each gene in the set of selected genes.

**Cross-sample gene variability.** The variability of a gene contrasts with how stable a gene is across samples. The idea of weighting genes based on their variability across samples is first explored by Wang et al.[22], wherein the variability was defined as the cross-sample variance. By weighting down the high variability genes, the authors achieved a great advantage over the traditional unweighted method. Genes with a low cross-sample variability better represent the sample population, hence are more trustworthy for learning the cell composition. AdRoit incorporates the same notion to weigh the importance of genes; however, it defines the variability more comprehensively. AdRoit acknowledges the dependency between the gene expression level and its variance and thus computes a variance-to-mean ratio (VMR) as the cross-sample variability. Here the mean and the variance are computed across samples. The distribution of the VMR values is less skewed than that of the variance alone. The VMR also circumvents underweighting genes with a low expression, or overweighting genes highly dispersed. Let $Y_{ij}$ denote the counts for gene $i$ in sample $j \in 1, \dots, J$, then

$$Y_{ij} \sim NB(\lambda_{ij}, p_{ij}), \tag{8}$$

where $\lambda_{ij}$ is the dispersion parameter of the gene $i$ in sample $j$, and $p_{ij}$ is the success probability. Again, we use MLE to estimate $\widehat{\lambda_{ij}}$ and $\widehat{p_{ij}}$, following which the cross-sample mean and variance can be numerically computed:

$$\mu_i^S = \frac{\widehat{\lambda_{ij}} \cdot \widehat{p_{ij}}}{1 - \widehat{p_{ij}}}, \tag{9}$$

$$(\sigma_i^2)^S = \frac{\widehat{\lambda_{ij}} \cdot \widehat{p_{ij}}}{(1 - \widehat{p_{ij}})^2}, \tag{10}$$

and the cross-sample variability VMR for gene $i$ is then defined as

$$VMR_i = \frac{(\sigma_i^2)^S}{\mu_i^S} = \frac{1}{w_i^C}, \tag{11}$$

where $w_i^C$ is later used in the model. The cross-sample variability weight is computed for each gene in the set of selected genes.

Typically, the RNA-Seq data of mixed cells to be deconvoluted include multiple replicated samples with similar cellular compositions. In this case, the VMR is computed as described above using the cross-sample mean and variance of the gene. In case the RNA-Seq data to be deconvoluted lack multiple replications whereas the single-cell reference contains replicates, AdRoit pools cells of the same type in each replicate to synthesize multiple samples to compute the gene means and variances per cell type, and subsequently estimate the VMRs with them. If neither input has multiple repeats, AdRoit takes a bootstrapping approach to sample and pool cells from each cell type in the single-cell reference several times to generate multiple samples.

**Gene-wise correction for the platform bias.** When comparing the RNA-Seq data of the mixture samples to the single-cell data, one needs to account for the possible library size and platform difference. The simplest means is to adopt a fixed global correction for each sample, e.g., all genes in a sample are linearly transformed in the same way. This operation is based on a strong assumption that the platform difference impacts every gene equally and is linearly scalable among different cell types, which hardly holds. In addition, because linear models are sensitive to outliers, the estimation of cell proportions can be steered away from the truth by genes that are largely affected by the platform bias. Therefore, applying a uniform correction to all genes is inappropriate.

To overcome this problem, AdRoit estimates a gene-wise correction per sample via an adaptive learning strategy and applies it to each gene respectively. To proceed, we first input the mean expression of gene $i$ from the replicated mixture samples ($\mu_i$) and the estimated means of each cell type from the single-cell data ($\mu_{ik}$ in Eq. (4)), then obtain a coarse estimation (e.g., via a non-negative least square regression[57]) of the proportions of each cell type, $\tau_k$. For each gene $i$, a predicted mean expression ($\sum_k^K \widehat{\tau_k}\mu_{ik}$ in Eq. (13)) in the sample of mixed cells is computed as the weighted sum of the means of this gene in each cell type wherein the weights are $\tau_k$. The regression equation is given by,

$$\mu_i = A \cdot \left(\sum_k^K \tau_k \mu_{ik} + \varepsilon\right), 0 < \tau_k, \sum_k^K \tau_k = 1 \tag{12}$$

where $A$ is a constant to ensure the sum of $\tau_k$'s is 1 and $\varepsilon$ is the error term. Next, we calculate the ratio between the mean expression from the mixture samples and the predicted counterpart, which constitutes the platform bias. We define the correction factor as a function that depends on the estimated platform bias. An option is the logarithm of the ratio plus 1,

$$r_i = \log_2\left(\frac{\mu_i}{\sum_k^K \widehat{\tau_k}\mu_{ik}} + 1\right) \tag{13}$$

Given the dispersed nature of count data, the logarithm results in relatively more stable values. The addition of 1 avoids taking logarithm on zeros. By applying the gene-wise $r_i$, the genes affected more strongly by the platform bias (i.e., "outliers") receive a larger correction (Fig. 1b).

**Weighted and regularized loss function for training.** AdRoit employs a non-negative least square regression model to infer the cell compositions in a compound RNA-Seq sample of mixed cell types. The model (Eq. (14)) approximates the gene expression observed in the compound sample by the weighted (the weights are the mixing percentages) sum of the cell type-specific gene expressions from the single-cell reference, after correcting the platform difference.

$$y_{ij} \approx r_i \cdot \sum_k^K \beta_k \widehat{\mu_{ik}} \, (\beta_1, \dots, \beta_K > 0) \tag{14}$$

Where $y_{ij}$ is the expression of gene $i$ in the compound transcriptome sample $j$; the coefficient $\beta_k$ is the mixing proportion for cell type $k$; $\widehat{\mu_{ik}}$ is the estimated mean expression of gene $i$ in cell type $k$ computed according to Eq. (4); $r_i$ is the gene-wise correction computed by Eq. (13) to alleviate the bias due to the technology difference. To infer the $\beta_k$ coefficients, AdRoit uniquely incorporates all the factors discussed in the previous sections. This is implemented in a weighted sum-of-squared loss function $L$, where the weights consist of two components, $w_i^C$ in Eq. (7) and $w_i^S$ in Eq. (11). Thus, a larger weight is given to the genes with a higher cell-type specificity and a lower cross-sample variability. In cases of complex tissues where many similar cell subtypes exist, strong collinearity among the subtypes can render the model sensitive to noise and prone to overfitting. AdRoit handles this problem by including an L2 norm of the mixing proportions as the regularization component in the loss function. For a compound transcriptome sample $j$, the loss function is given by,

$$L_j\left(\beta_1, \dots, \beta_K|y_{ij}, w_i^C, w_i^S, r_i, \widehat{\mu_{ik}}\right) = \sum_i^I w_i^C \cdot w_i^S \cdot \left(y_{ij} - r_i \cdot \sum_k^K \beta_k \widehat{\mu_{ik}}\right)^2 + \lambda \sum_k^K \beta_k^2 \tag{15}$$

where $i$ is the index of a gene among the selected genes used for the deconvolution; $y_{ij}, \beta_k, r_i$, and $\widehat{\mu_{ik}}$ are the same as defined in Eq. (14); $w_i^S$ and $w_i^C$ are computed as described above in Eqs. (7) and (11), respectively. $\lambda$ is the strength of the regularization. The $\beta_k$ coefficients are estimated by minimizing the loss function

subject to the constraint $\beta_1, \ldots, \beta_K > 0$,

$$\widehat{\beta_1}, \ldots, \widehat{\beta_K} = \underset{\beta_1, \ldots, \beta_K > 0}{\arg\max} \, L_j \tag{16}$$

The minimization is done by a gradient projection method proposed by Byrd et al.[58]. We derive the gradient function by taking the partial derivative of the loss function with respect to $\beta_k$,

$$G_k = \nabla_{\beta_k} L_j = -2 \sum_i^I r_i \cdot \widehat{\mu_{ik}} \cdot w_i^C \cdot w_i^S \cdot \left( y_{ij} - r_i \cdot \sum_k^K \beta_k \widehat{\mu_{ik}} \right) + 2\lambda \beta_k \tag{17}$$

AdRoit uses the function 'optim' from the R package 'stats' to perform the minimization[59], providing the loss function (Eq. (15)) and the gradient (Eq. (17)). To get the final estimates of cell-type proportions, we rescale the coefficients $\beta_k$'s to ensure a summation of 1,

$$\theta_k = \frac{\widehat{\beta_k}}{\sum_k^K \widehat{\beta_k}} \tag{18}$$

Each compound RNA-Seq sample $j$ is independently estimated by the model described above.

**Evaluation statistics**. We compared the estimated cell-type proportions with the ground truth by calculating the following four statistics. The mean absolute difference (mAD) and root mean square deviation (RMSD) are given by,

$$mAD = \frac{\sum_k^K |\theta_k - \theta_k^0|}{K} \tag{19}$$

$$RMSD = \frac{\sum_k^K (\theta_k - \theta_k^0)^2}{K} \tag{20}$$

where $\theta_k$ and $\theta_k^0$ are the estimated proportions and true proportions, respectively.
    Pearson correlation coefficient is computed as,

$$\rho_p = \frac{\sum_k^K (\theta_k - \overline{\theta_k})(\theta_k^0 - \overline{\theta_k^0})}{\sqrt{\sum_k^K (\theta_k - \overline{\theta_k})^2} \sqrt{\sum_k^K (\theta_k^0 - \overline{\theta_k^0})^2}} \tag{21}$$

where $\overline{\theta_k}$ and $\overline{\theta_k^0}$ are the means of the estimated proportions and true proportions, respectively. Spearman correlation coefficient is given by,

$$\rho_s = \frac{\sum_k^K (r_k - \overline{r_k})(r_k^0 - \overline{r_k^0})}{\sqrt{\sum_k^K (r_k - \overline{r_k})^2} \sqrt{\sum_k^K (r_k^0 - \overline{r_k^0})^2}} \tag{22}$$

where $r_k$ and $r_k^0$ is the rank of $\theta_k$ and $\theta_k^0$, respectively, and $\overline{r_k}$ and $\overline{r_k^0}$ are the means of $r_k$'s and $r_k^0$'s.

**Generation of synthesized bulk RNA-Seq and spatial transcriptomics data**.
Bulk RNA-Seq data used for the benchmarking were synthesized by adding up the raw UMI counts per gene from all cells within a subject. Let $t_k$ denote a cell of cell type $k$, $t_k \in 1, \ldots, T_k$, where $T_k$ is the total number of cells in cell type $k$. Let $Y_{ij}^B$ be the count of gene $i$ in a synthesized bulk sample associated with the subject $j$, and $X_{ijt_k}$ be the UMI count of gene $i$ in cell $t_k$ within the subject $j$, then

$$Y_{ij}^B = \sum_k^K \sum_{t_k}^{T_k} X_{ijt_k} \tag{23}$$

The true proportion of cell type $k$ is given by,

$$\theta_k^0 = \frac{T_k}{\sum_k^K T_k} \tag{24}$$

    To simulate the spatial transcriptomic spots, we first sampled 10 cells without replacement from each cell type, then weighed them by their respective mixing proportions before adding them up. For example, to simulate a spot with $p_k$ percent of cell type $k$, the count $Y_{ij}^s$ of gene $i$ in a spatial spot $j$ is given by,

$$Y_{ij}^s = \sum_k^K p_k \sum_{n=1}^{10} X_{ikn} \tag{25}$$

where $X_{ikn}$ is the UMI count of gene $i$ in a selected cell $n$ of cell type $k$. For each mixing scheme, the simulation was repeated 100 times.

**Single-cell RNA sequencing of mouse dorsal root ganglion**. As described previously[60], lumbar DRGs were isolated from five 18-week old adult C57BL/6 mice (three males and two females) and transferred to a dissociation buffer (Dulbecco's modified Eagle's medium supplemented with 10% heat-inactivated Fetal Calf Serum) (Gibco; cat # A38400-02). To generate a single cell suspension, DRGs were subjected to a 2 step-enzymatic dissociation followed by a mechanical dissociation. In brief, DRGs were first incubated with 0.125% collagenase P from Clostridium histolyticum (Roche Applied Science; cat # 11249002001) for 90 min in an Eppendorf Thermomixer C (37 °C; intermittent 750 rpm shaking for about 10 s every 2 min). Then, DRGs were transferred to a Hank's Balanced Salt Solution

(HBSS, $Mg^{2+}$ and $Ca^{2+}$ free; Invitrogen) supplemented with 0.25% Trypsin (Worthington biochemical corp.; cat # LS003707) and 0.0025% EDTA and incubated for 10 min at 37 °C in the Eppendorf Thermomixer C. Trypsin was neutralized by the addition of 2.5 mg/ml MgSO4 (Sigma; cat #M-3937) and DRGs were triturated with Pasteur pipettes. The resulting cell suspension was passed through a 70 μm mesh filter to remove remaining chunks of tissues and centrifuged for 5 min at 2500 rpm at room temperature. The pellet was resuspended in HBSS ($Ca^{2+}$, $Mg^{2+}$ free; Invitrogen) and the cell suspension was run on a 30% Percoll Plus gradient (Sigma GE17-5445-02) to further remove debris. Finally, cells were resuspended in PBS supplemented with 0.04% BSA at a concentration of 200 cells/μl and cell viability was determined using the automated cell analyzer NucleoCounter® NC-250™. The suspended single cells were loaded on a Chromium Single Cell Instrument (10x Genomics) with about 6000 cells per lane to minimize the presence of doublets. 2000–3000 cells per lane were recovered. RNA-Seq libraries were constructed using Chromium Single Cell 3′ Library, Gel Beads & Multiplex Kit (10x Genomics). Single-end sequencing was performed on Illumina NextSeq500. Read 1 starts with a 26 bp UMI and cell barcode, followed by an 8 bp i7 sample index. Read 2 contains a 55 bp transcript read. Sample de-multiplexing, alignment, filtering, and UMI counting were conducted using Cell Ranger Single-Cell Software Suite[61] (10x Genomics, v2.0.0). Mouse mm10 Genome assembly and UCSC gene model were used for the alignment.

**Data preprocessing**. For the DRG single-cell data, the UMI data output from the Cell Ranger Single-Cell Software Suite (10x Genomics, v2.0.0) was analyzed using Seurat package[62] to assess the cell quality and identify cell types, similar to what was described previously[42]. Cells with the number of detected genes less than 500 or over 15000, or with a UMI ratio of mitochondria encoded genes versus all genes over 0.1 were also removed. The UMI data was normalized by the 'NormalizeData' method in Seurat with default settings. To avoid potential sample-to-sample variation caused by the technical variation at various experiment steps, we employed Seurat data integration method. The top 2000 variable genes of each of the 5 DRG samples were identified using 'FindVariableFeatures' with selection.method = 'vst'. Based on the union of these variable genes, the anchor cells in each sample were identified by 'FindIntegrationAnchors'. All the samples were then integrated by 'IntegrateData'. We subsequently scaled the integrated data ('ScaleData') and performed dimension reduction ('RunPCA'). Cells were then clustered based on the first 15 principal components by applying 'FindNeighbors' and 'FindClusters' (resolution = 0.6, algorithm = 1). Marker genes for each cluster were identified using 'FindAllMarkers'. Parameters were used such that these genes were expressed in at least 25% of the cells in the cluster, and on average two-fold higher than the rest of cells with a multiple-testing adjusted Wilcoxon test $p$ value of less than 0.01. The specificity of the canonical cell-type-specific genes or cell cluster-specific genes were further examined by visualizations (Supplementary Fig. 4) and used to define the cell type for each cluster. In the end, the original UMI data from 17271 genes and 3352 cells that passed the quality control were organized into a matrix (genes as rows and cell identifiers as columns). This matrix, together with the cell type label for each cell therein, were loaded into AdRoit as the reference profiles.
    To build the mouse brain single-cell atlas, the scRNA-Seq reference data of the mouse brain were obtained from Zeisel et al.[32]. Among all the available data, we only retained 96,572 cells that were acquired from the brain regions, had an assigned cell type by the authors, and a minimal total UMI of 1000. These cells correspond to 183 clusters at the finest taxonomy level in the original study. As many of the clusters are highly similar, we decided to merge some of them to simplify the reference landscape. First, the top 50 cluster enriched markers were derived using Scanpy[63] via the 'rank_genes_groups' function (method = 'wilcoxon'), following the normalization ('normalize_per_cell'), log transformation ('log1p') and regressing out ('regress_out') the variances associated with the total UMI and the percentage of mitochondrial chromosome encoded genes per cell. Then, the pair-wise overlapping p-values among the clusters were calculated using the top 50 marker genes assuming a hypergeometric null distribution. Last, clusters with overlapping p-values more significant than 1e−10 were merged and new names were assigned by combinedly considering the original annotation, the molecular features, and the specificity to certain brain regions. A total of 46 cell types were determined that cover all the 12 brain regions and their important substructures[40] (Supplementary Data 5). To make the reference dataset more manageable in size and more balanced in the representation of cell types, we down-sampled each cluster to no more than 360 cells. A final set of 14,666 cells over 46 cell types was used for the deconvolution of the mouse brain spatial transcriptome data.
    The human Islets single-cell RNA-Seq data contained 1492 high-quality human islets single cells and the associated annotations from Xin et al.[41]. The RPKM expression table was directly downloaded and used as-is. The RNA-FISH data was also from this study[41]. The real bulk RNA-Seq data of human islets was acquired from a large-scale study referred in multiple publications[41,43,44]. We only included the data from donors with a valid HbA1C level measurement in the regression analysis of the Beta-cell proportion with respect to the HbA1C level (Fig. 5c and Supplementary Data 10).
    The human trabecular meshwork single-cell data were acquired from the authors of Patel et al.[42]. We used the UMI counts, tSNE coordinates, and the cell type annotation of 8759 high-quality cells. Details of the quality control and the cell type identification were presented in the original publication.

The Mouse Brain Spatial transcriptomics data by 10x Visium platform were download from the 10x Genomics website (see "Data availability"), including the filtered count matrix, tissue image, and the spatial coordinates of a coronal section of an adult C57BL/6 mouse brain. A total of 2698 in-tissue spots were provided in the data set and used as-is.

The Mouse Brian ISH images were directly downloaded from Allen mouse Brain Atlas[40] by searching the gene names. The images were used without further editing except for cropping.

**Statistics and reproducibility**. The statistical analyses were done with R statistical software (v3.6.0)[59] and python (v3.7.2)[64]. The packages used include Seurat (v3.0.1)[62], scanpy (v1.6.0)[63], dplyr (v0.8.0.1)[65], doParallel (v1.0.14)[66], data.table (v1.12.4)[67], fitdistrplus (v1.1-1)[56], nnls (v1.4)[57], MuSiC (v0.1.1)[22], BisqueRNA (v1.0.4)[21], SPOTlight (v1.0.4)[38], Cell2location (v0.05-alpha)[39], Stereoscope(v_03)[23].

**Reporting summary**. Further information on research design is available in the Nature Research Reporting Summary linked to this article.

# Data availability

The mouse DRG single-cell data were deposited at NCBI GEO (accession number: GSE163252). The bulk RNA-Seq and RNA-FISH data for human pancreatic islets were initially published as aggregated data where the data processing and experimental procedure were described therein[41,43,44]. We acquired the individual sample data from the authors and released them along with the current study (Supplementary Data 10). The other public data analyzed in this study were obtained using GEO accession number GSE81608 (human pancreatic islets single-cell data), NCBI SRA accession number PRJNA616025 (human trabecular meshwork single-cell data), and NCBI SRA accession number SRP135960 (mouse brain single-cell data). The 10x Genomics PBMC data[68] and the Visium mouse brain spatial transcriptomics data[69] were downloaded from 10x Genomics website.

# Code availability

AdRoit's source code is available on Github (https://github.com/TaoYang-dev/AdRoit)[70].

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

## Acknowledgements

We thank Yurong Xin for pointing us to the relevant public data resource. We also thank Gabor Halasz and Yuan Zhu for the helpful discussions regarding the memory usage, and Shawn Mishra for the manuscript proofreading.

## Author contributions

Y.B., T.Y., N.A.-H., M.L.-F., L.E.M., and G.S.A. designed the research. T.Y., Y.B., and W.F. developed the algorithm. T.Y., Y.B., and J.K. participated in the data analysis. M.S. and R.B. performed the DRG tissue collection. C.A. performed the single-cell library preparation and sequencing experiment. Y.B., T.Y., and N.A.-H. wrote the manuscript.

## Competing interests

T.Y., Y.B., W.F., and G.S.A. have filed a patent application relating to the AdRoit computational framework. M.L.-F. is an employee of Cellular Longevity. The remaining authors are employees and shareholders of Regeneron Pharmaceuticals, although the manuscript's subject matter does not have any relationship to any products or services of this corporation.
