## [Transparent Peer Review File · Communications Biology]

Reviewers' comments:

Reviewer #1 (Remarks to the Author):

In this manuscript 'AdRoit: an accurate and robust method to infer complex transcriptome composition' Yang et al., employed an adaptive learning approach for cell type deconvolution and shows implication of this method in analysing spatial transcriptomics data. Overall, this is an important area of research to combine retrospective bulk RNA-seq data with upcoming scRNA-seq studies. However, given the presence of multiple methods in the area of cell type deconvolution authors needs incorporate more granularity in the analysis.

I have three major suggestions before manuscript can be accepted for publication.

- 1) Granularity: Given the presence of multiple methods in this area new methods should bring more granularity in cell deconvolution. For example, whether AdRoit can deconvolute different lymphoid (CD4, CD8, NK, NKT etc) or myeloid (Monocyte, macrophages or DCs) in bulk data?
- 2) Sensitivity: Quantification of cell types in complex tissue is very challenging problem and authors employed approaches (6 out of the 12 cell types) which lacks the complexity of tissue (~25-30 cells) as well as validation by in situ approaches. These claims are not substantiated by considering complexity of tissue and/or validation.
- 3) Benchmarking: Author should also including methods such as SPOTLIGHT and Cell2location for bench-marking AdRoit.

Reviewer #2 (Remarks to the Author):

Yang et al. developed AdRoit, a method for deconvoluting bulk RNA-seq and spatial transcriptomics data. My major comments are below.

Various parts of the paper can be improved for better clarity. This is especially important in the abstract. For example, the authors mentioned "compound RNA-seq" but only introduced this term later in Introduction. Also, when talking about cross-platform applicability, it would be good to be specific (i.e., the technical difference between single-cell and "compound" RNA-seq data).

The authors only benchmarked MuSiC and NNLS. While I appreciate the authors' efforts to adopt the various validation strategies, some key methods need to be benchmarked against to support the authors' claims. For example, CIBERSORT-x and Bisque both address the differences between bulk RNA-seq and scRNA-seq, and at least one should be included in the comparison when the authors emphasize the cross-platform correction of AdRoit. Also, there are several methods developed for spatial deconvolution, and some are easy to run to be included in the benchmark study. At least, they should be referenced.

How does AdRoit utilize the spatial coordinates in the spatial transcriptomic data?

For the regularization term in the deconvolution step, how to interpret the beta coefficients? How are the estimated proportions scaled / biologically meaningful after this l1 penalty? More importantly, how to carry out inference?

Response To Reviewers

Reviewer # 1

General: In this manuscript 'AdRoit: an accurate and robust method to infer complex transcriptome composition' Yang et al., employed an adaptive learning approach for cell type deconvolution and shows implication of this method in analysing spatial transcriptomics data. Overall, this is an important area of research to combine retrospective bulk RNA-seq data with upcoming scRNA-seq studies. However, given the presence of multiple methods in the area of cell type deconvolution authors needs incorporate more granularity in the analysis.

Comment 1: Granularity: Given the presence of multiple methods in this area new methods should bring more granularity in cell deconvolution. For example, whether AdRoit can deconvolute different lymphoid (CD4, CD8, NK, NKT etc) or myeloid (Monocyte, macrophages or DCs) in bulk data?

Our response:

To assess the level of granularity that AdRoit can achieve when deconvoluting similar cell types, we added one additional benchmarking analysis using cell mixtures that contain closely related immune cells from myeloid or lymphoid lineage, as the reviewer suggested. Specifically, based on a public data set of human peripheral blood mononuclear cell (PBMC) single cells from 10x Genomics, we synthesized bulk samples that consist of naïve CD4+ T cells, memory CD4+ T cells, and CD8+ T cells at various mixing proportions, as well as samples that contain CD14+ monocytes, FCGR3A+ monocytes, and dendritic cells at different percentage compositions. By comparing the AdRoit-inferred cell proportions with the ground truth, we demonstrated that AdRoit can deconvolute closely resembled cell types with high accuracy. The results were presented in Fig. 3a (the updated Fig. 3 is included below), Supplementary Table 5, and lines 196-212 in the revised manuscript (please refer to the line numbers after all changes are accepted).

Additionally, in our original submission, we presented the performance of AdRoit in deconvoluting highly homologous neuron subtypes in mouse dorsal root ganglia. The cell mixtures we used contain five peptidergic neuron subtypes, three non-peptidergic neuron subtypes, and three neurofilament-containing neuron subtypes. The leading accuracy we observed, compared to the other three methods, provides an independent confirmation that AdRoit offers a high level of granularity. This analysis remains in the revised manuscript (Fig. 4 and lines 264-293; please refer to the line numbers after all changes are accepted).

Comment 2: Sensitivity: Quantification of cell types in complex tissue is very challenging problem and authors employed approaches (6 out of the 12 cell types) which lacks the complexity of tissue (~25-30 cells) as well as validation by in situ approaches. These claims are not substantiated by considering complexity of tissue and/or validation.

Our Response:

We agree with the reviewer that this is an important point to address. To help evaluate the performance of AdRoit in complex tissues, we benchmarked AdRoit and other methods using a new set of synthetic bulk samples of high complexity. The data source for creating this benchmarking dataset is a published mouse brain single cell atlas that contains 46 major brain cell types after consolidation (Methods, lines 799-816 in the revised manuscript; please refer to the line numbers after all changes are accepted). Every sample was generated by pooling cells from 30 different cell types that were randomly

selected from the total 46 cell types. This procedure was repeated independently 100 times for thorough coverage of various cell type combinations. By a head-to-head comparison in the scatterplot (Fig. 3d), we observed that AdRoit was able to estimate the proportions closely consistent with the ground truth while the other tools made less satisfactory predictions. Further, the receiver operating characteristic (ROC) curve indicates that AdRoit has the best sensitivity and specificity given its highest area-under-the-curve (AUC) (Fig. 3e, AdRoit: 0.99, Bisque: 0.77, MuSiC: 0.80, SPOTlight: 0.67). These results were summarized in Fig. 3d, e and lines 248-261 (please refer to the line numbers after all changes are accepted).

We agree with the reviewer that *in situ* protein or RNA detection methods, for instance, immunohistochemistry or RNAscope/RNA-FISH, can be used to detect and quantify cell types. However, many of these methods are based on the visualization of fluorescence. Overlapping wavelength spectra of currently available fluorochromes limit the maximum number of probes, making simultaneous detection of 25-30 cell types of a complex tissue impractical using standard protocols. Besides, these *in situ* methods are often slide-based, and the tissue content on the slide may not represent the whole tissue used in the bulk RNA-seq experiments. Despite these limitations, we agree with the reviewer that it is a good idea to use ISH, when suitable, as an orthogonal approach to confirm the results. For example, the human islet is predominated by 4 cell types. Our collaborators were able to dissociate the islet tissue to make the slide for RNA-FISH and quantified the cell types using four fluorophores. We utilized this set of RNA-FISH data to validate our deconvolution results in the bulk samples of the matching subjects (Fig. 6b). Moreover, *in situ* methods can be used for the evaluation of spatial transcriptome deconvolution results for specific cell types. We precisely applied ISH for this purpose in the manuscript (Fig. 6d).

Comment 3: Benchmarking: Author should also include methods such as SPOTLIGHT and Cell2location for bench-marking AdRoit.

Our Response:

We thank the reviewer for suggesting additional methods to enrich our analysis. SPOTlight has claimed that it is suitable for both bulk RNA-seq and spatial transcriptome, whereas Cell2location only declared its application in the spatial transcriptome data. Thus, in the revised manuscript, we added SPOTlight and another recently emerged method, Bisque, to all the benchmarking datasets regarding the deconvolution of bulk RNA-seq data, and SPOTlight and Cell2location to all the deconvolution analyses of the spatial transcriptome data. We noticed that AdRoit remained the most favorable among all tested methods. Results are presented in Figures 2-5 (see updated figures below) and the accompanying text (lines 168-370; please refer to the line numbers after all changes are accepted), supplementary figures 1,3,5-7 and supplementary tables 2,4,5,6,7,8,10-12.

Reviewer # 2

General: Yang et al. developed AdRoit, a method for deconvoluting bulk RNA-seq and spatial transcriptomics data. My major comments are below.

Various parts of the paper can be improved for better clarity. This is especially important in the abstract. For example, the authors mentioned “compound RNA-seq” but only introduced this term later in Introduction. Also, when talking about cross-platform applicability, it would be good to be specific (i.e., the technical difference between single-cell and “compound” RNA-seq data).

Comment 1: Various parts of the paper can be improved for better clarity.

Our response:

We thank the reviewer for pointing this out. We modified various parts of the manuscript, including the abstract, to improve the clarity. For instance, the “compound RNA-seq” is no longer referenced prior to its definition, and the “cross-platform applicability” is further explained. Changes are tracked in the revised manuscript.

Comment 2: The authors only benchmarked MuSiC and NNLS. While I appreciate the authors’ efforts to adopt the various validation strategies, some key methods need to be benchmarked against to support the authors’ claims. For example, CIBERSORT-x and Bisque both address the differences between bulk RNA-seq and scRNA-seq, and at least one should be included in the comparison when the authors emphasize the cross-platform correction of AdRoit. Also, there are several methods developed for spatial deconvolution, and some are easy to run to be included in the benchmark study. At least, they should be referenced.

Our response:

We thank the reviewer for raising this important issue and suggesting additional methods for consideration. We expanded our benchmarking analyses to include three more deconvolution tools. In particular, we added SPOTlight and Bisque to all the benchmarking datasets regarding the deconvolution of bulk RNA-seq data, and SPOTlight and Cell2location to all the deconvolution analyses of the spatial transcriptome data. SPOTlight was considered for both tasks because it was considered by the authors to be suitable for both bulk RNA-seq and spatial transcriptome. In the expanded analysis, AdRoit remained the most favorable among all tested methods. Results are presented in Figures 2-5 (see updated figures below) and the accompanying text (lines 168-370; please refer to the line numbers after all changes are accepted), supplementary figures 1,3,5-7 and supplementary tables 2,4,5,6,7,8,10-12.

As our affiliated organization is for-profit, we are not permitted to use CIBERSORT-x according to its *Term of Use* statement (“Use of the Service by any commercial entity for any purpose, including research, is prohibited”). Even though we cannot do a direct comparison, several tools evaluated in our work (e.g. SPOTlight, Bisque) have done so and their results suggested that CIBERSORT-x is not superior compared to these methods. Therefore, we expect AdRoit remains the lead against CIBERSORT-x.

Comment 3: How does AdRoit utilize the spatial coordinates in the spatial transcriptomic data?

Our response:

Spatial coordinates were not used in AdRoit in the scope of this work. Each spatial spot was considered as an independent sample when conducting the deconvolution. We only used the coordinates to visualize the inferred cell proportions.

Comment 4: For the regularization term in the deconvolution step, how to interpret the beta coefficients? How are the estimated proportions scaled / biologically meaningful after this l1 penalty? More importantly, how to carry out inference?

Our response:

As the reviewer mentioned, the regularization term in the loss function is an L2 norm, also known as the sum of squares, of the β_k coefficients ($\sum_k^K \beta_k^2$ in eq. 15). The coefficient β_k , after a

rescaling to ensure their sum is 1 (see below), represents the mixing proportion for cell type k . Using these coefficients to formulate the regularization term does not affect this interpretation of the β_k 's in our model. It's worth noting that the deconvolution model and the loss function used to train the model are different entities. Our model approximates the expression observed in the bulk sample by solely the weighted (weights are the β_k 's) sum of the cell type specific expressions (offered by the single cell reference data). The L2 term is only used in the loss function, together with the term associated with the difference between the observation and the approximated value by the model ($\sum_i^I w_i^C \cdot w_i^S \cdot (y_{ij} - r_i \cdot \sum_k^K \beta_k \widehat{\mu}_{ik})^2$ in eq. 15). The purpose of the regularization is to prevent overfitting. In the revised manuscript, we modified the relevant text for better clarity (lines 678-710; please refer to the line numbers after all changes are accepted).

The β_k estimates are rescaled simply by dividing each β_k by the sum of all β_k 's to guarantee they add up to 1. The rescaled β_k values represent the percentage of cell types and sum to 100%, which is biologically meaningful. Again, the L2 norm of the β_k 's is part of the loss function not the model, so it does not interfere with the fact that the β_k 's are defined as the mixing percentages and sum up to 1.

As for the inference, we used a gradient projection method, where we first derived the gradient function by taking the derivative of the loss function with respect to each β_k . Given the loss function and the derived gradient, we used the 'optim' function from the R package 'stats' to minimize the loss function. The detailed inference was described in lines 703-710 of the revised manuscript (please refer to the line numbers after all changes are accepted).

Updates in the Figures:

Fig. 1: No change

Fig. 2: Kept the initial Fig. 2a and 2b; added Bisque and SPOTlight in the comparison.

Fig. 2

Fig. 3: Added Fig. 3a, 3d, and 3e; 3b and 3c were taken from the initial Fig. 2c and 2d, respectively; added Bisque and SPOTlight in the comparison.

Fig. 3

Fig. 4: Renamed from the initial Fig. 3; Added Bisque and SPOTlight and removed NNLS in Fig. 4b and 4c.

Fig. 4

Fig. 5: Renamed from the initial Fig. 4; Added Cell2location and SPOTlight in 5a, 5b and 5c.

Fig. 5

Fig. 6: Renamed from the initial Fig. 5 with no change

REVIEWERS' COMMENTS:

Reviewer #1 (Remarks to the Author):

Authors have address all my concerns in revised version. I recommend the acceptance of this manuscript.

Reviewer #2 (Remarks to the Author):

The authors have addressed my previous concerns, and I appreciate their efforts in carrying out the additional benchmark results.

For the L2 penalty in the loss function, there is a large literature on "selective inference." The beta coefficients have various degrees of freedom depending on the model size and their standard errors are hard to be assessed.